# Pulmonary Adverse Events in Cancer Immunotherapy: Case Studies of CT Patterns

**DOI:** 10.3390/diagnostics14060613

**Published:** 2024-03-14

**Authors:** Giorgio Bocchini, Maria Chiara Imperato, Tullio Valente, Salvatore Guarino, Roberta Lieto, Candida Massimo, Emanuele Muto, Federica Romano, Mariano Scaglione, Giacomo Sica, Davide Vitagliano Torre, Salvatore Masala, Marialuisa Bocchino, Gaetano Rea

**Affiliations:** 1Department of Radiology, Monaldi Hospital, Azienda Ospedaliera dei Colli, 80131 Naples, Italy; tullio.valente@gmail.com (T.V.); salvatore.guarino@ospedalideicolli.it (S.G.); roblieto@gmail.com (R.L.); candida.massimo@ospedalideicolli.it (C.M.); emanuele.muto@ospedalideicolli.it (E.M.); federica.romano@ospedalideicolli.it (F.R.); gsica@sirm.org (G.S.); gaetano.rea71@gmail.com (G.R.); 2Department of Radiology, S. M. I. dell’Olmo Hospital, Azienda Ospedaliera Universitaria San Giovanni di Dio e Ruggi D’Aragona, 84013 Cava de’ Tirreni, Italy; mcimperato@gmail.com; 3Radiology Department of Surgery, Medicine and Pharmacy, University of Sassari, 07100 Sassari, Italy; mscaglione@tiscali.it (M.S.); samasala@uniss.it (S.M.); 4Department of Radiology, James Cook University Hospital & Teesside University, Middlesbrough TS4 3BW, UK; 5Radiology Department, Pineta Grande Hospital, 81030 Castel Volturno, Italy; 6Department of Radiology, S. Francesco D’Assisi Hospital, 84020 Oliveto Citra, Italy; davide.vitaglianotorre@gmail.com; 7Department of Clinical Medicine and Surgery, Section of Respiratory Diseases, University Federico II, Monaldi Hospital, Azienda Ospedaliera dei Colli, 80131 Naples, Italy; marialuisa.bocchino@gmail.com

**Keywords:** immune-related adverse events, immune-checkpoint inhibitors, lung toxicity, organizing pneumonia, non-specific interstitial pneumonia, hypersensitivity pneumonitis, sarcoid-like reaction

## Abstract

Immune-checkpoint inhibitors have profoundly changed cancer treatment, improving the prognosis of many oncologic patients. However, despite the good efficacy of these drugs, their mechanism of action, which involves the activation of the immune system, can lead to immune-related adverse events, which may affect almost all organs. Pulmonary adverse events are relatively common, and potentially life-threatening complications may occur. The diagnosis is challenging due to the wide and non-specific spectrum of clinical and radiological manifestations. The role of the radiologist is to recognize and diagnose pulmonary immune-related adverse events, possibly even in the early stages, to estimate their extent and guide patients’ management.

Immune-checkpoint inhibitors (ICIs) are a class of immunotherapy drugs designed to block specific immune checkpoints, proteins that are involved in regulatory pathways of the immune system and are crucial to avoid autoimmunity, and to limit tissue damage in response to pathogens; nevertheless, they can be exploited by tumor cells as mechanisms of immunoevasion and immunoresistance [1].

ICIs currently used in clinical practice are anti-cytotoxic T lymphocyte-associated antigen-4 (CTLA-4) and anti-programmed cell death protein 1/programmed cell death ligand 1 (PD-1/PD-L1) monoclonal antibodies [2]. 

Approved ICIs with their clinical indications [1,2] are listed in Table 1.

The frequency of immune-related adverse events (irAEs) varies according to the immune-checkpoint target, but generally, the risk and severity are higher with CTLA-4 inhibitors than with PD-1/PD-L1 inhibitors, and even higher when ICIs are combined with other molecules [3].

Otherwise, pulmonary irAEs are more common in patients receiving PD-1/PD-L1 inhibitors rather than CTLA-4 inhibitors alone [4,5].

The onset and severity of pulmonary irAEs are unpredictable and variable from patient to patient; previous lung radiotherapy, asthma and/or smoking may predispose patients to develop higher-grade ICI-associated pneumonitis [6]. Although the physiopathology of irAEs remains to be fully understood, it is presumed that ICIs may overstimulate the immune system and alter host homeostasis, causing an excessive inflammatory response [7].

Clinical manifestations include dry cough, dyspnea, hypoxia, tachycardia, fever and chest pain, but patients may also be asymptomatic [8,9,10].

The median time for pneumonitis development is 2–3 months from the beginning of therapy. The incidence of ICI-induced pneumonitis is higher in combined therapy (6.5–10%) than in monotherapy (3–4%) [10].

Computed Tomography (CT) imaging plays a fundamental role in the diagnosis of pulmonary irAEs, suggesting whether ICI therapy should be suspended and support therapies should be introduced [9,10].

The main high-resolution chest CT (HRCT) patterns identified in patients treated with ICIs are organizing pneumonia (OP), non-specific interstitial pneumonia (NSIP), hypersensitivity pneumonitis (HP), acute interstitial pneumonia/acute respiratory distress syndrome (AIP/ARDS) and obliterative bronchiolitis (OB); among these patterns, OP is the most common [9,10,11,12]. 

Lung toxicity can manifest even with a sarcoid-like reaction (SLR) involving pulmonary parenchyma and mediastinal and hilar lymph nodes; it occurs in 5–7% of patients treated with ICIs, with a median time to onset of 6 months [11,13]. 

Recently, acute eosinophilic pneumonia (AEP) has been reported as a pulmonary irAE in a few case reports [12,14,15], but the incidence of this lung pattern could have been underestimated and classified as OP since they have similar findings on chest CT [15].

Radiologic patterns of lung toxicity are summarized in Table 2.


**Patterns of presentation**


The aim of the article is to provide updated information on the most frequent CT patterns of presentation of pulmonary toxicity from immunotherapy applied to cancer patients, with some examples from our clinical practice, to keep radiologists informed in this constantly evolving field.

Some examples of the listed patterns are depicted in Figure 1, Figure 2, Figure 3, Figure 4, Figure 5, Figure 6 and Figure 7.

(**A**–**D**) A 64-year-old man with a diagnosis of small-cell lung cancer with adrenal and brain metastases, treated with nivolumab. Six months after beginning immunotherapy, the patient presented progressive dyspnea and dry cough. Chest auscultation revealed diffuse crackles associated with a moderately restrictive functional pattern. The blood count did not show significant leukocytosis but showed increased inflammatory indexes. The patient benefited from a temporary suspension of immunological therapy with steroid intake, but the subsequent relapse forced a radical change in therapeutic strategy. 

Axial chest CT images (**A**–**D**) show bilateral confluent areas of GGOs and peripheral and peribronchovascular consolidative opacities.

The structural alteration of the right pulmonary hilum due to the presence of neoplastic tissue, with the infiltration of the ipsilateral main bronchus (**A**) and the “reversed halo sign” (white arrow (**D**)), are also visible.

**Figure 2 diagnostics-14-00613-f002:**
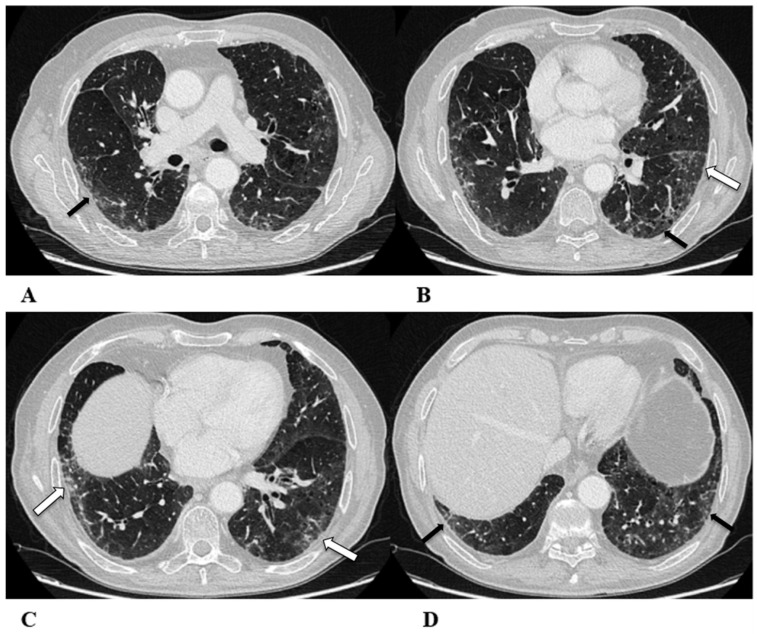
Non-specific interstitial pneumonia (NSIP). Non-specific interstitial pneumonia represents the second most commonly described pattern of ICI-related pneumonitis. It is defined pathologically by different degrees of interstitial inflammatory cell infiltrates and/or fibrosis. HRCT characteristics in patients with NSIP consist of GGOs associated with irregular reticulations and traction bronchiectasis/bronchiolectasis. Honeycombing is infrequent and minimally represented. NSIP distribution is usually bilateral and symmetric, predominantly subpleural and basal, and often with sparing of the immediate subpleural lung.

The NSIP pattern should be distinguished from atypical pneumonia based on clinical analysis. The greater involvement of the lung bases and recurrent relative subpleural sparing must be considered in the clinical assessment of the patient, as they are uncommon findings in infectious processes. It is also necessary to exclude the concomitance of immune disorders and connective tissue diseases [9,12].

(**A**–**D**) A 70-year-old man with lung adenocarcinoma and brain metastases treated with pembrolizumab, carboplatin and pemetrexed as first-line treatment. Two months later, the patient had dyspnea and dry cough, without fever.

Axial chest CT scans show mild subpleural reticulation in the lower lobes (black arrows (**A**,**B**,**D**)) and patchy bilateral and partially confluent areas of GGO in both lungs predominant in the lower lobes (white arrows (**B**,**C**)). Incidental findings: rare small lucent areas of centrilobular emphysema, especially in both upper lobes.

**Figure 3 diagnostics-14-00613-f003:**
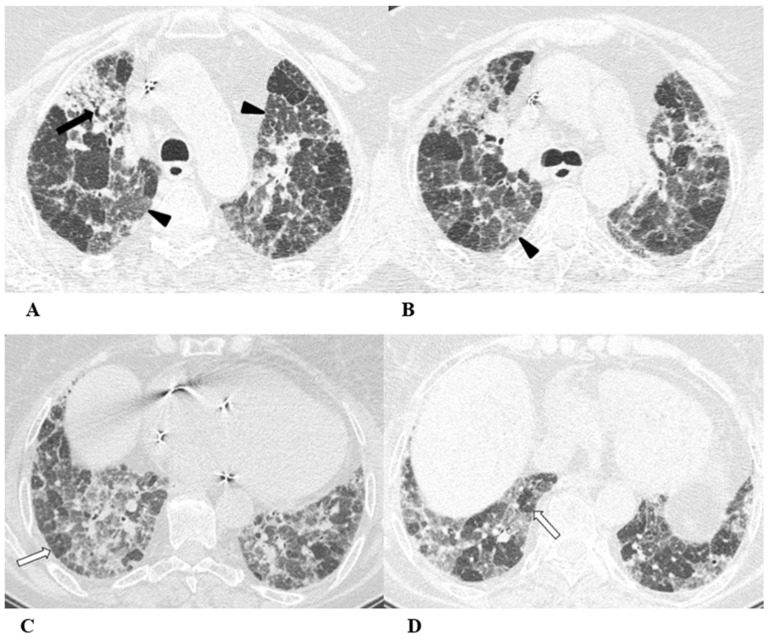
*Hypersensitivity pneumonia (HP)*. Hypersensitivity pneumonia is characterized by lymphocyte infiltrates that thicken the alveolar septa, with poorly formed non-necrotizing granulomas and multinucleated giant cells.

Typical HRCT features of acute HP are multiple bilateral small centrilobular nodules with upper lung zone predominance, lobular areas of decreased attenuation and vascularity suggestive of “air trapping” and patchy and/or diffuse GGO areas [9,12,17]. Chronic HP shows a typical “three density pattern” characterized by bilateral and diffuse lobular areas of decreased attenuation (air-trapping areas), patchy areas of real GGO in a predominant peribronchovascular distribution with traction bronchiectasis and bronchiolectasis inside (fibrosis signs), and also areas of a relative increase in density related to the normal CT appearance of the lung (GGO-like areas). 

The ICI-associated HP pattern is indistinguishable from that of HP induced by allergen exposure. For this reason, in addition to the exclusion of atypical pneumonia, accurate exposure history is essential for the correct evaluation of the patient.

(**A**–**D**) A 76-year-old woman with colorectal cancer who received nivolumab and ipilimumab as second-line treatment. Seven months later, the patient presented worsening cough and dyspnea, along with weight loss, but she was able to undergo HRCT examination just two months after the onset of symptoms.

Axial chest CT scans of the upper lobes (**A**,**B**) show patchy GGO areas (black arrowheads) with over-imposed reticulation in a predominant peribronchovascular distribution and early traction bronchiectasis (black arrow).

Bilateral areas of decreased attenuation (air trapping) (white arrows) with geometric morphology are easily recognizable in both the inspiratory (**A**,**B**) and expiratory (**C**,**D**) acquisitions.

**Figure 4 diagnostics-14-00613-f004:**
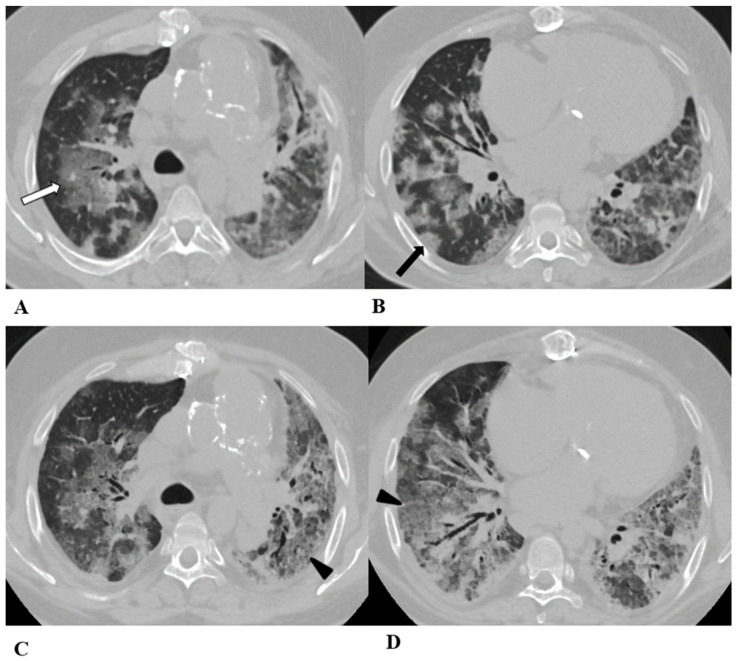
*Acute interstitial pneumonia (AIP)/acute respiratory distress syndrome (ARDS).* Acute interstitial pneumonia (AIP) and acute respiratory distress syndrome (ARDS) can be a consequence of both pathogenic intrapulmonary and extrapulmonary stimuli. They are clinically characterized by acute respiratory failure, a consequence of alveolar thickening due to hyaline membrane deposition and inflammatory cells’ infiltration. Both conditions likely represent the same pathology, with AIP probably accounting for some of the idiopathic cases of ARDS.

ARDS from extrapulmonary disease shows bilateral symmetrical changes on CT, whereas in pulmonary ARDS, the opacities tend to be asymmetrical.

The initial form of ARDS can also be classified as diffuse alveolar damage (DAD), followed by the organizing and delayed phases.

HRCT features depend on the phase of the disease, with the possibility of GGOs in the acute phase, and lung cysts and pulmonary opacifications often with an anteroposterior gradient in the delayed phase; complete resolution or evolution into a coarse reticular pattern, traction bronchiectasis and possible bullae, as result of prolonged ventilation, may be found. Sometimes, pleural effusion is observed [9,12,16]. 

(**A**–**D**) A 45-year-old man with rectal carcinoma metastatic to the liver, treated with pembrolizumab. After nearly three months of immunotherapy, the patient suffered acute respiratory failure and interstitial pneumonia suggestive of ARDS in the clinical setting. 

Baseline axial chest CT scans (**A**,**B**) showed numerous partially confluent areas of GGO, more represented in the perihilar regions as a result of extensive alveolar damage (*white arrow*), with some dependent asymmetrical peripheral consolidations (*black arrow*) like an early ARDS pattern, which made it necessary to hospitalize the patient in the intensive respiratory care unit with nose–tracheal intubation. 

Three weeks later, CT images (**C**,**D**) demonstrated lung architecture disruption, a large component of GGO, reticulation with areas of crazy-paving (*black arrowheads*), incremented consolidation in the dependent lung and traction bronchiectasis, as seen in the delayed fibrotic ARDS phase.

As accessory findings, note the altered profile of the right ventricle outflow tract (RVOT) and pulmonary trunk with wall calcifications, as a result of multiple reconstructive operations in pediatric age for tetralogy of Fallot.

**Figure 5 diagnostics-14-00613-f005:**
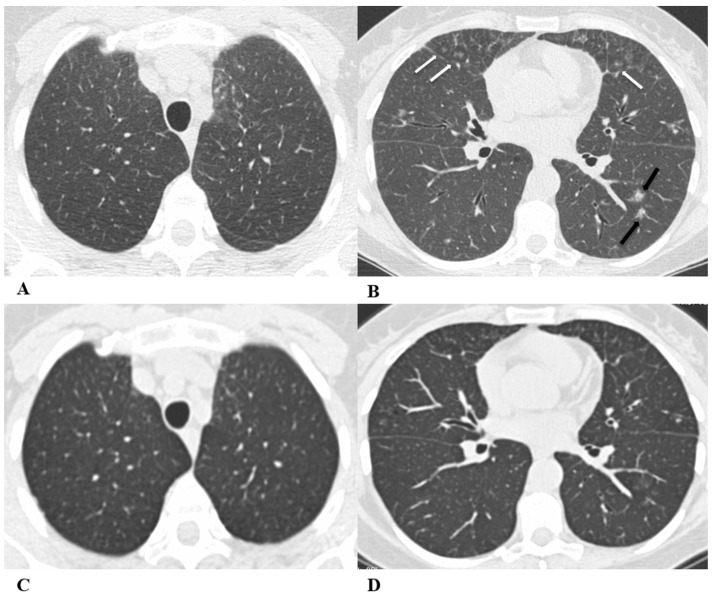
*Obliterative bronchiolitis (OB)*. Obliterative bronchiolitis, also known as bronchiolitis obliterans, is an uncommon lung manifestation associated with ICI therapy. OB represents a type of bronchiolitis characterized by the concentric luminal narrowing of the small airways due to submucosal and peribronchiolar inflammation and fibrosis. HRCT scans, especially several weeks after the acute onset, can show the presence of sharply defined areas of decreased lung attenuation with reduced caliber vessels, giving a mosaic attenuation pattern, better seen on expiratory scans, if necessary. Bronchial wall thickening and centrilobular opacities are common, while advanced findings such as reticulations and traction bronchiectasis are poorly described because of the early diagnosis of drug-induced toxicity [18,19].

In the past, the bronchiolitis model was not considered a manifestation of therapeutic toxicity due to its similar appearance to the infectious and inflammatory causes of bronchiolitis. However, suspicion of this condition as an ICI-related pattern should arise when no infectious symptoms are present and should be confirmed through imaging; it is necessary to demonstrate the resolution of the pathological findings either by suspending ICIs or by demonstrating this after treatment with steroids.

Moreover, to differentiate bronchiolitis from other causes of infection, there are further features to consider; aspiration pneumonia typically occurs in the dependent lung regions with fluid stagnation in the bronchi lumen and in the presence of esophageal ingestions, while other infective pneumonias are often clinically distinguishable.

(**A**–**D**) A 57-year-old man with gastric cancer and splenic metastasis treated with pembrolizumab. Cough and moderate dyspnea affected the patient two months after the beginning of immunotherapy. 

Axial chest CT scans demonstrate, in the paramediastinal region of the left upper lobe, subtle thickening of the bronchiolar walls associated with an area of modest ground-glass with rare centrilobular micronodules (**A**) and bilateral thickening of the bronchial walls (**B**). In the middle lobe and lingula, there is the presence of multiple small centrilobular opacities with initial thickening of the interlobular septa, which better defines the anatomy of some contiguous secondary lobules (*white arrows* (**B**)). Bilaterally, especially in the lower left lobe, there is the presence of other patchy areas of pulmonary opacity similar to those found in other inflammatory–infectious processes (*black arrows* (**B**)).

The chest CT performed 20 days after pembrolizumab interruption showed a marked improvement in lung condition (**C**,**D**).

**Figure 6 diagnostics-14-00613-f006:**
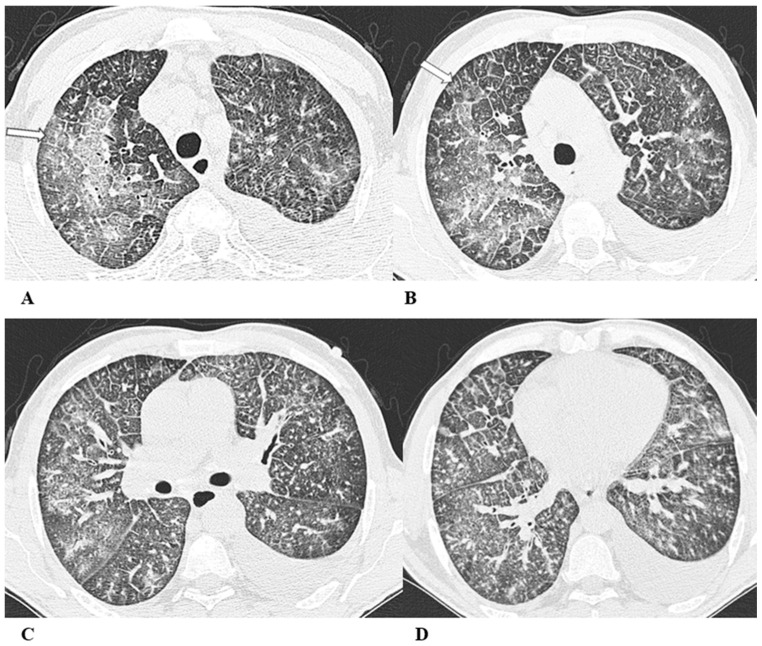
*Acute eosinophilic pneumonia (AEP).* Acute eosinophilic pneumonia is defined by the diffuse infiltration of eosinophils and lymphocytes within the alveoli and interstitium; sometimes, interstitial fibrosis is observed.

Differential diagnosis with atypical pneumonias is necessary. Blood tests demonstrate that peripheral eosinophilia and immunoglobulin E values may be slightly elevated. When performed, thoracentesis may reveal a pleural effusion with eosinophilia.

On HRCT, the AEP pattern is characterized by bilateral patchy areas of GGO, often associated with peribronchovascular and septal thickening, especially in the upper lobes [12,14,15,20]. Moderate pleural effusion and areas of consolidation or poorly defined nodules may be observed [20].

(**A**–**D**) A 77-year-old man with renal cell carcinoma and previous micronodular lung metastases in treatment with nivolumab. After four months, he developed fatigue, cough and tachypnea, but no fever. The eosinophil count in the peripheral blood (812 cells/μL) obviated the need for bronchoalveolar lavage (BAL). Eosinophilia was not observed before the ICI therapy.

Axial chest CT images show diffuse and confluent GGO areas in both lungs associated with smooth septal thickening (crazy-paving pattern) (*white arrows*
**A**,**B**); micronodules with random distribution, attributable to micro-metastases, are predominant in both lower lobes (**C**,**D**). Pleural effusion is visible on the left side.

**Figure 7 diagnostics-14-00613-f007:**
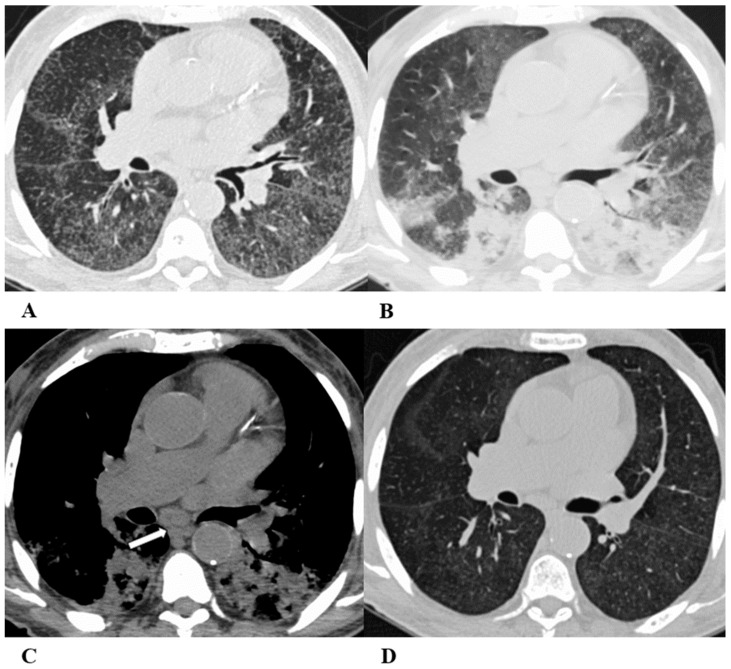
*Sarcoid-like reaction (SLR).* A sarcoid-like reaction is a less-common non-pneumonitis irAE determined by the presence of non-caseating granulomatous inflammation in the lung parenchyma and lymph nodes, without criteria for systemic sarcoidosis [21].

Sarcoid-like reaction HRCT findings are represented in most cases by bilateral and symmetrical hilar and mediastinal lymphadenopathy, sometimes associated with peribronchovascular micronodules, peribronchial interstitial thickening and the infrequent coalescence of micronodules forming pseudo-masses with a predominantly dependent distribution. Lung parenchyma may be involved without adenopathy [11,13,22]. 

Bronchoalveolar lavage (BAL) shows moderate lymphocytic alveolitis, and its first-line treatment is with oral corticosteroids.

(**A**–**D**) A 75-year-old man with myelodysplasia and renal cell carcinoma. Fever, fatigue, cough and dyspnea affected the patient from the fifth month of treatment with pembrolizumab.

The axial chest CT image showed widespread micronodules with a miliary and perilymphatic distribution (**A**). Six days later, along with a moderate clinical worsening, the chest CT highlighted the development of large confluent areas in both lower lobes, with peripheral and declivous locations and the persistence of diffuse bilateral parenchymal micronodules (**B**); a mediastinal window of the same CT scan of image B demonstrated the volumetric enlargement of some lymphadenopathies in the sub-carinal station (*white arrow* (**C**)).

Chest CT performed 20 days later (**D**) after the discontinuation of pembrolizumab and treatment with oral steroids showed significant improvement in the pulmonary picture, with the complete resolution of the consolidations and a tendency towards the resolution of the bilateral parenchymal micronodulia.

Other thoracic findings not consistent with typical drug-induced pneumonia are described in the literature, including radiation recall pneumonia and transient asymptomatic pulmonary opacities (TAPOs).

Radiation recall pneumonitis (RRP) refers to inflammation in previously irradiated lung tissue following exposure to a triggering pharmacological agent (such as anticancer medications like chemotherapy, targeted therapy and immunotherapy) and presents with new pulmonary ground-glass and/or consolidative opacities months or even years after completing radiation therapy that aligns with the radiation treatment plan [23,24].

TAPOs are characterized as asymptomatic, localized, and spontaneously resolving infiltrates that may display chest CT patterns of organizing pneumonia, simple eosinophilic pneumonia, and nodular morphology [23].

Another uncommon immunotherapy-induced lung disease is represented by pneumonia flare [25].

The increasing use of ICIs in cancer treatment has significantly improved the survival rate of several oncologic patients [26], but it has also widened the variety of drug-induced lung diseases (DILDs).

Recent studies report that the development of irAEs is associated with increased tumor response to ICI therapy in patients with advanced cancers [27,28,29]. 

ICI-related pulmonary manifestations show a wide spectrum of appearances and severity ranging from serious and rapid worsening respiratory symptoms associated with a chest CT pattern of AIP/ARDS, requiring admission to the intensive care unit, to mild–moderate severity with dyspnea, cough and radiological patterns of OP, NSIP, HP, OB, SLR or AEP that benefit from corticosteroid treatment.

Based on clinical symptoms, pulmonary irAEs are currently classified by the Common Criteria for Adverse Event Terminology (CTCAE) into five categories, ranging from the absence of symptoms with only radiological changes to acute respiratory compromise and death [30].

The extent of lung CT manifestations related to ICIs can be quantitatively assessed with scoring systems such as the Royal Marsden Hospital (RMH) DILD score by associating to each lobe a score of zero, one or two depending, respectively, on whether there is the absence of lung manifestations, less than 50% of affected parenchyma or more than 50% of affected parenchyma. The RMH DILD score has shown a reliable correlation with the CTCAE grades, and it may represent an additional tool for clinical and therapeutic evaluation [31]. 

The use of CT-based semiquantitative scores of lung involvement in pulmonary irAE, such as the RMH DILD score, although not yet widely used, is advisable as it is fast and readily available to guide more effective and rapid treatment strategies, and other semiquantitative chest CT scores have shown to be correlated with the patient’s prognosis and have proven to be particularly valuable even in the assessment of interstitial and infective lung diseases [32,33].

Therefore, in patients treated with immunotherapy, imaging plays a crucial role not only in assessing tumor response but also in evaluating adverse events, even when patients are asymptomatic, as early diagnosis can prevent the worsening of clinical conditions.

Unfortunately, pulmonary irAE diagnosis is often challenging, as symptoms and imaging features are not specific, mimicking tumor progression and other lung diseases like infections or radiation pneumonitis, thus delaying appropriate patient management [31]. Because no definitive test is available, they often represent a diagnosis of exclusion, taking into account the temporal relationship between ICI administration and the onset of clinical or radiologic presentation.

When irAE manifests as consolidation, especially in the case of the OP pattern, it could be misinterpreted as tumor progression [34,35]. A lack of lesion progression in other organs and the regression of the consolidations after corticosteroid therapy help in the differential diagnosis.

Differential diagnosis with infectious pneumonia may be verified with a nasal swab, sputum culture and bronchioalveolar lavage; imaging contribution is based on the typical CT patterns, pointing out that on chest CT, bacterial pneumonia usually shows consolidations with air-bronchograms and enlarged hilar lymph nodes associated with pleural effusion [36], while viral pneumonia may present with diffuse bilateral GGOs and consolidation areas or centrilobular nodules, occasionally with pleural effusion [37].

A clinical history of lung radiation therapy may suggest radiation pneumonitis when GGO areas evolve in well-defined consolidations (not limited by interlobar fissures or by broncho-vascular structures), linear scarring and traction bronchiectasis associated with lobar volume loss and adjacent pleural thickening or effusion [38]. 

In the aforementioned complex radiologic scenario, relevant support could be provided by the emerging and innovative research field of radiomics. A few recent studies [39,40,41] show encouraging results on the accuracy of radiomics models, based on data extracted from chest CT images, in differentiating lung irAEs from other lung diseases. 

In conclusion, pulmonary irAEs represent potential and not uncommon adverse effects of immunotherapy in cancer patients undergoing ICI therapy. 

Radiologists must therefore be familiar not only with imaging response criteria to immunotherapy but also with the different patterns of lung irAE. Furthermore, given that the extent of pulmonary involvement reflects the severity of the disease, radiological evaluation plays a critical role in therapeutic decisions.

## Figures and Tables

**Figure 1 diagnostics-14-00613-f001:**
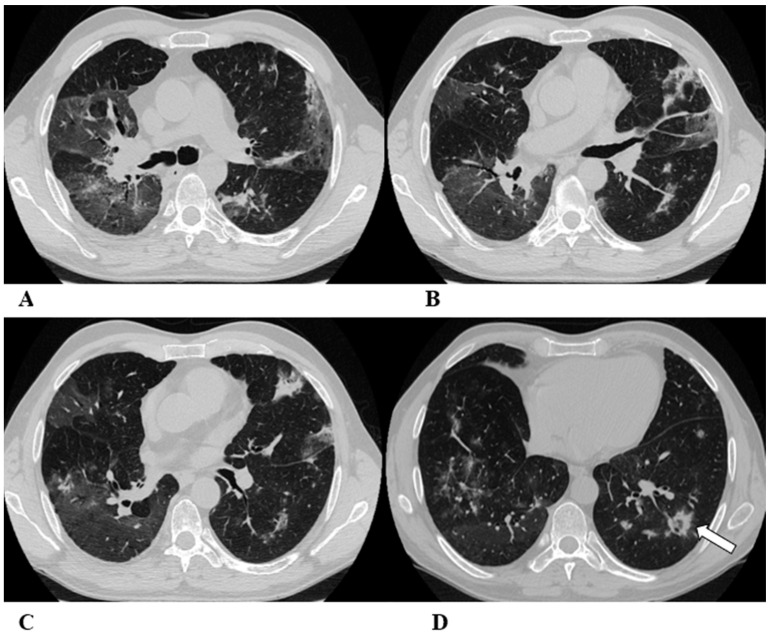
Organizing pneumonia (OP). Organizing pneumonia is determined by the presence of granulation tissue buds in the alveoli, alveolar ducts and distal bronchioles. On HRCT, OP is characterized by diffuse and bilateral areas of consolidation and ground-glass opacities (GGOs), often migratory, with predominantly peripheral/subpleural and peribronchovascular distribution [9,12]. Other OP findings are the “reversed halo sign” (a focal area of GGO surrounded by a ring of denser air-space consolidation; if the surrounding ring is incomplete, it is mentioned as the “atoll sign” [16]) and, rarely, centrilobular nodules.

**Table 1 diagnostics-14-00613-t001:** United States Food and Drug Administration (US FDA), European Medicines Agency (EMA) and China National Medical Products Administration (NMPA)-approved immune-checkpoint inhibitors’ indications.

Immune-Checkpoint Inhibitor	Targets	US FDA/EMA-Approved Indications	China NMPA-Approved Indications
Pembrolizumab	PD-1	Melanoma, non-small-cell lung cancer, head and neck cancer, Hodgkin’s lymphoma, urothelial carcinoma, MSI-H/dMMR colorectal cancer, gastric cancer, cervical cancer, hepatocellular carcinoma, Merkel cell carcinoma, renal cell carcinoma, small-cell lung cancer, esophageal carcinoma, endometrial cancer, myelodysplasiadata	Melanoma, non-small-cell lung cancer, head and neck cancer, MSI-H/dMMR colorectal cancer, esophageal squamous cancer, hepatocellular carcinoma
Nivolumab	PD-1	Melanoma, non-small-cell lung cancer, renal cell carcinoma, Hodgkin’s lymphoma, head and neck cancer, urothelial carcinoma, MSI-H/dMMR colorectal cancer, hepatocellular carcinoma, small-cell lung cancer	Non-small-cell lung cancer, head and neck cancer, esophageal squamous cancer, gastric cancer
Cemiplimab	PD-1	Cutaneous squamous cell carcinoma	
Toripalimab	PD-1	Nasopharyngeal carcinoma	Melanoma, nasopharyngeal carcinoma
Sintilimab	PD-1		Hodgkin’s lymphoma, non-small-cell lung cancer, esophageal cancer, hepatocellular carcinoma, gastric cancer
Camrelizumab	PD-1		Hodgkin’s lymphoma, esophageal squamous carcinoma, Non-small-cell lung cancer, hepatocellular carcinoma, nasopharyngeal carcinoma
Tislelizumab	PD-1	Non-small-cell lung cancer, esophageal squamous carcinoma	Hodgkin’s lymphoma, non-small-cell lung cancer, esophageal squamous carcinoma, nasopharyngeal carcinoma hepatocellular carcinoma, uroepithelial carcinoma
Penpulimab	PD-1		Hodgkin’s lymphoma, non-small-cell lung cancer, nasopharyngeal carcinoma
Zimberelimab	PD-1		Hodgkin’s lymphoma, cervical cancer
Serplulimab	PD-1		Gastric cancer, non-small-cell lung cancer, MSI-H/dMMR colorectal cancer
Pucotenlimab	PD-1		MSI-H/dMMR colorectal cancer, melanoma
Atezolizumab	PD-L1	Urothelial cancer, non-small-cell lung cancer, breast cancer, small-cell lung cancer	Non-small-cell lung cancer
Durvalumab	PD-L1	Urothelial carcinoma, non-small-cell lung cancer	
Avelumab	PD-L1	Merkel cell carcinoma, urothelial carcinoma, renal cell carcinoma	
Envafolimab	PD-L1		MSI-H/dMMR colorectal cancer
Sugemalimab	PD-L1		Non-small-cell lung cancer, esophageal squamous carcinoma
Tremelimumab	CTLA-4	Melanoma, non-small-cell lung cancer, renal cell carcinoma, hepatocellular carcinoma	
Ipilimumab	CTLA-4	Melanoma, metastatic renal cell carcinoma, MSI-H/dMMR colorectal cancer	
Cadonilimab	PD-1/CTLA-4		Cervical cancer

**Table 2 diagnostics-14-00613-t002:** Classification of the most frequent immune-checkpoint inhibitors (ICIs) in lung injury CT patterns.

PNEUMONITIS	
Organizing pneumonia (OP)	Multifocal areas of consolidation and ground-glass opacities (GGOs) with peribronchovascular and subpleural distribution; occasionally, reversed halo signs and perilobular patterns are visible.
Non-specific interstitial pneumonia (NSIP)	Bilateral patchy areas of GGO, irregular reticular opacities and traction bronchiectasis with predominant lower-lung involvement.
Hypersensitivity pneumonitis (HP)	Bilateral lobular areas of air trapping, patchy GGO areas and poorly defined centrilobular nodules; typical basal sparing.
Acute interstitial pneumonia/acute respiratory distress syndrome (AIP- ARDS)	Extensive and confluent bilateral areas of GGO and/or consolidation, especially in the dependent lung.
Obliterative bronchiolitis (OB)	Bronchial wall thickening, occasionally bronchiectasis and centrilobular nodules are present. Areas of mosaic attenuation can be present.
Eosinophilic pneumonia (EP)	Bilateral and symmetrical hilar and mediastinal lymphadenopathies, sometimes associated with peribronchovascular micronodules and peribronchial interstitial thickening.
**SARCOID-LIKE REACTIONSarcoid-like reaction** (**SLR**)	Bilateral and symmetrical hilar and mediastinal lymphadenopathies, sometimes associated with peribronchovascular micronodules and peribronchial interstitial thickening.

## Data Availability

Not applicable.

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
