# Peer review of "Pulmonary Adverse Events in Cancer Immunotherapy: Case Studies of CT Patterns"

_diagnostics, 2024, doi:10.3390/diagnostics14060613_

Round 1

Reviewer 1 Report

Comments and Suggestions for Authors

Comments:

This review article by Bocchini et al. provides a comprehensive overview of pulmonary adverse events associated with immune checkpoint inhibitor (ICI) treatment, with a focus on CT findings. The authors rightly highlight the crucial role of CT in the differential diagnosis, classification of patterns, and assessment of severity in ICI-related pneumonitis, a topic extensively discussed in previous review articles and original research. While the content of this review generally follows existing literature and provides illustrative examples from the authors' clinical experience, it lacks novelty in terms of presenting new information or alternative interpretations. The figures and tables, depicting typical cases along with their clinical course, have educational value. However, given the limited clinical content beyond imaging, referencing other relevant reviews or incorporating additional clinical perspectives could enhance the comprehensiveness of this review. Several concerns were identified by the reviewer that necessitate revision in the following areas:

  1. Update on Clinical Information:

    • In Table 1, the absence of FDA-approved Tremelimumab raises questions. Additionally, considering the approval of numerous new agents in China (Int J Cancer. 2023; 152(11): 2351-2361.), a revision incorporating the latest list of items is recommended. The reviewer also suggests adding at least one oncologist as a co-author to enhance the description's clinical context.
    • Line 57, Admitting the frequency of immune-related adverse events (irAEs) associated with CTLA-4 inhibitors is generally greater than PD-(L)1 inihibitors, particularly in pulmonary complications, this is not the case. It is also noted the increased risk with combination therapy. Referencing other clinical reviews, such as (Respir Investig. 2020 Sep;58(5):305-319), would provide additional insights.
    • Regarding eosinophilic pneumonia (EP), clarification is sought on whether the presented case aligns with the "simple pulmonary eosinophilia pattern" previously defined in the Fleischner position paper. If so, the appropriateness of Figure 6 as a typical example is questioned, given the presence of numerous lung metastases and interpretational challenges regarding septal thickening. Further explanation regarding eosinophil infiltration, response to steroids, and typical EP progression is warranted.
  2. Lack of Evidence in Conclusion:

    • The recommendation for using CT-based semiquantitative scores, such as the RMH DILD score, lacks sufficient supporting evidence. While the authors cite a single reference using DLID score in L283, validation of correlation with CTCAE grade as well as predictive value of clinical outcomes are unclear. A more extensive discussion regarding the merits of semiquantitative analysis of CT scans beyond RMH DILD score, supported by additional literature, is necessary to accept the conclusions.
  3. minor points; L294 'miming', should be 'mimicking' please double check.
Comments on the Quality of English Language

No problem.

Author Response

Our gratitude to the esteemed Reviewer for the insightful comments and valuable suggestions, which have facilitated the expansion of our paper and enhanced the quality of our case series.

Below, you will find our detailed responses to each comment, along with the corresponding revisions highlighted in the resubmitted file.

In Table 1, the absence of FDA-approved Tremelimumab raises questions. Additionally, considering the approval of numerous new agents in China (Int J Cancer. 2023; 152(11): 2351-2361.), a revision incorporating the latest list of items is recommended. The reviewer also suggests adding at least one oncologist as a co-author to enhance the description's clinical context. 

Table 1 has been revised.

MD Marialuisa Bocchino, one of the co-authors, is a Full Professor in Respiratory Medicine at the School of Medicine of the Federico II University of Naples. While her primary focus is on interstitial lung diseases, she has also overseen some oncology patients with pulmonary immune-related adverse events (irAEs) and has been involved in the clinical management of the cases presented in this paper. 

Line 57, Admitting the frequency of immune-related adverse events (irAEs) associated with CTLA-4 inhibitors is generally greater than PD-(L)1 inihibitors, particularly in pulmonary complications, this is not the case. It is also noted the increased risk with combination therapy. Referencing other clinical reviews, such as (Respir Investig. 2020 Sep;58(5):305-319), would provide additional insights. 

We have added the information and two new references.

Regarding eosinophilic pneumonia (EP), clarification is sought on whether the presented case aligns with the "simple pulmonary eosinophilia pattern" previously defined in the Fleischner position paper. If so, the appropriateness of Figure 6 as a typical example is questioned, given the presence of numerous lung metastases and interpretational challenges regarding septal thickening. Further explanation regarding eosinophil infiltration, response to steroids, and typical EP progression is warranted.

The patient depicted in Fig. 6 had shown pulmonary micrometastases before initiating therapy with Nivolumab, which subsequently disappeared during treatment. Approximately 5 months after starting therapy, pneumological symptoms emerged, and CT investigations revealed a typical pattern of Acute Eosinophilic Pneumonia (AEP) and Simple Pulmonary Eosinophilia (SPE). We believe that the case presented meets the criteria of SPE as briefly described in the Fleischner Society document of 2021 and in other interesting articles or reviews (e.g., Radiol Clin N Am 54, 2016, 1151–1164, doi: 10.1016/j.rcl.2016.05.008).

The pulmonary micronodules depicted in the figure, coupled with septal thickening and scattered areas of parenchymal thickening, could also represent another transient manifestation of drug-induced lung disease, as documented in the literature. In our assessment, the patient's condition was accurately classified, aided by the eosinophil count in the peripheral blood at that time (812 cells/μL), which obviated the need for Bronchoalveolar Lavage (BAL). Unfortunately, we do not have access to subsequent follow-up CT scans due to the patient's transfer; however, we were informed that the radiological abnormalities resolved approximately 1 month later, without discontinuation of Nivolumab but with the adjunct of steroid therapy.

Lack of Evidence in Conclusion:

The recommendation for using CT-based semiquantitative scores, such as the RMH DILD score, lacks sufficient supporting evidence. While the authors cite a single reference using DLID score in L283, validation of correlation with CTCAE grade as well as predictive value of clinical outcomes are unclear. A more extensive discussion regarding the merits of semiquantitative analysis of CT scans beyond RMH DILD score, supported by additional literature, is necessary to accept the conclusions.

Currently, there is not much evidence of the use of RMH DILD score for the evaluation of pulmonary irAEs. However, we advocate its use because, in our experience, it simplifies the decision-making process and enables more timely clinical and therapeutic evaluations; also considering that other chest CT semiquantitative scores have demonstrated their usefulness in different lung diseases, for example as widely demonstrated in the recent COVID-19 pandemic.

Clarifications have been added and moved to the beginning of the paragraph.

minor points; L294 'miming', should be 'mimicking' please double check.

Corrected.

Reviewer 2 Report

Comments and Suggestions for Authors

I don't think the paper brings any kind of novelty to the topic, beyond being a review, like many others.

Furthermore, there are no references to some rare, unusual adverse events, such as radiation recall pneumonia, pneumonia flare, transient asymptomatiyc pulmonary opacities, which could improve the completness of the paper.  

Author Response

Our gratitude to the esteemed Reviewer for the insightful comments and valuable suggestions, which have facilitated the expansion of our paper and enhanced the quality of our case series.

Below, you will find our detailed responses to each comment, along with the corresponding revisions highlighted in the resubmitted file.

Furthermore, there are no references to some rare, unusual adverse events, such as radiation recall pneumonia, pneumonia flare, transient asymptomatiyc pulmonary opacities, which could improve the completness of the paper.  

We have added the information and three new references.

Furthermore, there are no references to some rare, unusual adverse events, such as radiation recall pneumonia, pneumonia flare, transient asymptomatiyc pulmonary opacities, which could improve the completness of the paper.  

We have added the information and three new references.

Round 2

Reviewer 1 Report

Comments and Suggestions for Authors

The revised version addressed all the concerns raised by the present reviewer and significantly improved the accuracy of description.

Reviewer 2 Report

Comments and Suggestions for Authors

Thank you very much for providing a revised manuscript.

In my opinion, this is now a more comprehensive review on the topic